# Prevalence and drug resistance patterns of Gram-negative enteric bacterial pathogens from diarrheic patients in Ethiopia: A systematic review and meta-analysis

Achenef Melaku Beyene[1]*, Mucheye Gezachew[1], Desalegn Mengesha[2], Ahmed Yousef[3], Baye Gelaw[1]

1 Department of Medical Microbiology, College of Medicine and Health Sciences, University of Gondar, Gondar, Ethiopia, 2 Global One Health Initiative, East African Regional Office, Addis Ababa, Ethiopia, 3 Department of Food Science and Technology, Ohio State University, Ohio, Columbus, United States of America

* achenefmela@yahoo.com, tbeyene11@gmail.com

**Data Availability Statement:** All relevant data are within the paper and its Supporting Information files.

## Abstract

### Background

Diarrhoea is the leading cause of morbidity and mortality in the world particularly in developing countries and among vulnerable groups of the population. Gram-negative enteric bacterial pathogens (GNEBPs) are a group of organisms that reside mainly in the intestine and induce diarrhoea. Antimicrobial agents are usually the part of their treatment regimen. The therapeutic effect of antimicrobials is hindered by the emergence and spread of drug-resistant strains. The information regarding the prevalence and antimicrobial resistance patterns of GNEBPs in Ethiopia is limited and found in a scattered form.

### Objectives

This study was designed to determine the pooled prevalence and drug resistance patterns of GNEBPs by meta-analysis of data from diarrhoeic patients in Ethiopia.

### Method

A comprehensive literature search was conducted through internet searches using Google Scholar, PubMed, Science Direct, HINARI databases, and reference lists of previous studies. Published articles were included in the study based on priorly set inclusion and exclusion criteria. Results were presented in the forest plot, tables, and figures with a 95% confidence interval (CI). The inconsistency index ($I^2$) test statistics was used to assess heterogeneity across studies. The pooled prevalence estimate of GNEBPs and their drug resistance patterns were computed by a random-effects model. Software for Statistics and Data Science (STATA) version 14 statistical software was used for the analysis.

**Funding:** The authors received no specific funding for this work.

**Competing interests:** The authors have declared that no competing interests exist.

**Abbreviations:** GNEBPs, Gram-Negative Enteric Bacterial Pathogens; HIV, Human Immunodeficiency Virus; I², Inconsistency Index; KAP, Knowledge, Attitude and Practices; MESH, Medical Subject headlines; PRISMA, Preferred Reporting Items for Systematic Reviews and Meta-Analyses; SNNP, South Nation Nationalist People's region; UN, United Nations; USA, United State of America; WHO, World Health Organization.

## Result

After removing those articles which did not fulfil the inclusion criteria, 43 studies were included in the analysis. Studies were conducted in 8 regions of the country and most of the published articles were from the Amhara region (30.23%) followed by Oromia (18.60%) and Southern Nations, Nationalities, and Peoples' region (SNNP) (18.60%). The pooled prevalence of GNEBPs was 15.81% (CI = 13.33–18.29). The funnel plot indicated the presence of publication bias. The pooled prevalence of GNEBPs in Addis Ababa, Amhara, SNNP, and Oromia regions were 20.08, 16.67, 12.12, and 11.61%, respectively. The pooled prevalence was 14.91, 18.03, and 13.46% among studies conducted from 2006–2010, 2011–2015, and 2016–2021, respectively and it was the highest (20.35%) in children having age less than or equal to 15 years. The pooled prevalence of *Escherichia coli*, *Campylobacter* spp., *Shigella* spp., and *Salmonella enterica* were 19.79, 10.76, 6.24, and 5.06%, respectively. Large proportions (60–90%) of the isolates were resistant to ampicillin, amoxicillin, tetracycline, and trimethoprim-sulphamethoxazole. The pooled prevalence of multidrug resistance (MDR) was 70.56% (CI = 64.56–76.77%) and MDR in *Campylobacter* spp., *Shigella* spp., *E. coli*, and *S. enterica*. were 80.78, 79.08, 78.20, and 59.46%, respectively.

## Conclusion

The pooled estimate showed a high burden of GNEBPs infections and a high proportion of drug resistance characters to commonly used antimicrobial agents in Ethiopia. Therefore, performing drug susceptibility tests, establishing an antimicrobial surveillance system and confirmation by molecular techniques are needed.

## Introduction

World Health Organization (WHO) defines diarrhoea as the passage of three or more loose or liquid stools per day (24 hours). During diarrhoea, the water content and volume of stool and defecation frequency will usually increase. The syndrome may be accompanied by other illnesses like vomiting, fever, dysentery, nausea, and abdominal cramps. It is the leading cause of morbidity and mortality in the world and contributes about 4% of all deaths and 5% of health loss to disability [1–3]. Diarrhoea is the fifth leading cause of death and it contributes to one in nine deaths among children younger than 5 years [4,5]. The problem is severe among the vulnerable population such as children, people with HIV, the elderly, and other individuals having weak immunity. Many factors contribute to diarrhoea; however, childhood wasting (low weight-for-height score), unsafe water, and unsafe sanitation are the leading risk factors [5]. The incidence of diarrhoea is different among the regions or continents of the world. It is highly prevalent in Sub-Saharan Africa and South Asia. The report from WHO showed that these countries account for about 78% of all diarrheal deaths among children in the developing world [2,4]. Ethiopia is one of the top three countries with very high child mortality due to diarrhoea in Africa [5–7].

Diarrhoea can be induced by a variety of causes. However, infectious agents like viruses and bacteria are among the leading causes. Bacteria, particularly Gram-negative enteric bacterial pathogens (GNEBPs) are the common causes of the syndrome. The group includes bacteria that reside mainly in the intestine. Genera such as *Escherichia*, *Shigella*, *Campylobacter*, *Salmonella*, *Enterobacter*, *Klebsiella*, *Yersinia*, *Serratia*, *Proteus*, and others are included in the group.

However, the most common and significant pathogens are *S. enterica*, *E. coli*, *Campylobacter*, and *Shigella* spp. [8].

Antimicrobial agents are usually part of the treatment regimen, particularly on diarrhoea caused by bacteria. Due to the widespread and indiscriminate use of antimicrobials, several resistant strains are emerging which tend to spread globally [9,10]. Hence, antimicrobial resistance (AMR) is a global health threat and was recognized in the 2016 United Nations (UN) General Assembly [11]. It is one of the top challenges in achieving the 2030 UN sustainable development goals [10]. Infections caused by resistant organisms affect treatment outcomes, treatment costs, disease spread, and duration of illness, posing a challenge to the future of chemotherapy [12]. Some pathogenic strains are also developing resistance not only to one but to several agents, i.e., multidrug resistance [13,14].

Information regarding the prevalence and antimicrobial resistance patterns of GNEBPs in Ethiopia is limited, and available information is found in scattered forms. Hence, there is an interest to conduct a nationwide study. To fill this significant gap, this systemic review and meta-analysis was prepared. The review focused on the prevalence and antimicrobial resistance patterns of GNEBPs isolated from diarrheic patients in Ethiopia. The output of this systematic review and meta-analysis can be used by clinicians, policymakers, and researchers to make evidence-based decisions.

## Methods

### Literature search and selection

The published articles were searched based on preferred reporting items for systematic reviews and meta-analyses (PRISMA) guideline [15]. The search was performed from June to August 2021 using Google Scholar, PubMed, Science Direct, and HINARI databases. The search queries were set based on medical subject headlines (MESH) and Boolean logic. Relevant MeSH terms and keywords were used to retrieve all relevant articles from the databases listed above. The keywords and MeSH terms used were "enteric bacteria AND diarrhoea AND drug resistance AND Ethiopia", "*Salmonella* AND diarrhoea, AND Ethiopia AND drug resistance", "*Shigella* AND diarrhoea AND Ethiopia AND drug resistance", "*Escherichia coli* AND diarrhoea AND Ethiopia AND drug resistance", "*Campylobacter* AND diarrhoea AND Ethiopia AND drug resistance" "*Yersinia* AND diarrhoea AND Ethiopia AND drug resistance". Each bacterial genus was searched separately, and a search was also conducted on reference lists of previous studies to increase the chance of getting more articles. Only those articles which fulfil the selection criteria were used to analyse the information.

### Inclusion and exclusion criteria

Research conducted on GNEBPs from the diarrhoeic patient or their antimicrobial susceptibility in Ethiopia and full-length published articles in the English language were included in the analysis. To get updated information on the issue, articles published from 2010 to August 2021 were considered. Studies that did not focus on GNEBPs from the diarrhoeic patient or their antimicrobial susceptibility, anonymous reports, abstracts (incomplete information), and published articles before 2010 and unpublished information were not included in the study. Studies that were conducted to assess the knowledge, attitude, and practice (KAP) of the community or the professionals were not also included.

### Data extraction

The selected articles were coded, and data were collected using a format prepared in Microsoft Excel. The format consists of the author's name, study period, year of publication, study

design, study region, study population, sample size, sample type, age, gender, isolated bacteria species, their prevalence, resistance patterns of the isolates, and prevalence of multidrug resistance. The extracted data were checked at least twice for their accuracy.

## Quality control

The quality of eligible studies was checked using a set of criteria based on Joanna Briggs Institute critical appraisal tools including appropriateness of the research design to address the target population, adequate sample size, quality of paper, completeness of the information, and appropriateness of methods for isolation of the bacteria and appropriate statistical analysis [16]. The eligibility of selected articles was also assessed and approved by experts in the discipline.

## Data analysis

The data were compiled in Excel 2010 (Microsoft, Redmond, WA, USA) spreadsheet and summarized by descriptive statistics. A random-effect model was used to determine the pooled prevalence and the 95% confidence interval (CI). All statistical analysis was achieved by using Software for Statistics and Data Science (STATA; https://www.stata.com/company/our-sites/) version 14. The data were described using forest plots, figures, and tables. The presence of publication bias was assessed by funnel plot. Sub-group analysis was performed based on the regions in the country, age group; (children, adults, and all age groups), and year of study (2006–2010, 2011–2015, and 2016–2021). Statistical heterogeneity was evaluated by the inconsistency index ($I^2$) test. The $I^2$ provides an estimate of the percentage of the variability in effect estimates that is due to heterogeneity rather than sampling error or chance differences. Hence, the $I^2$ test measures the level of statistical heterogeneity among studies [17,18].

## Results

### Characteristics of published articles

Of 7,349 identified studies, 6,680 articles were excluded upon reviewing the titles and abstracts because they were irrelevant (were not focusing on GNEBPs, diarrhoea, and drug resistance or were outside Ethiopia or duplicates). The remaining 669 articles were assessed for eligibility; of these, 626 articles were excluded since they were review, KAP, or meta-analysis studies. Finally, 43 studies meeting the inclusion criteria were included in this study. Selected articles were focusing on one or more GNEBPs. Fig 1 shows a flow diagram of the selection of articles for the analysis.

Table 1 shows the overall characteristics of articles included in the analysis, type and prevalence of Gram-negative isolates recovered from diarrheic patients. Studies were conducted in 8 regions of the country and most of the published articles were from the Amhara region (30.23%) followed by the Oromia region (18.60%) and South Nation and Nationalities Region (SNNP) (18.60%). Published articles were not found in other regions of the country (Afar, Somali, and Benishangul Gumuz regions) (Fig 2).

All studies were cross-sectional; conducted from 2006 to 2020 and published online from 2010 to August 2021. Almost all studies (97.67%) were institution-based (conducted on patients visiting health facilities). The stool samples were the specimen used to isolate the bacterial species. Isolates were characterized and their identities were confirmed by cultural and conventional biochemical tests, but molecular techniques were not used. Patients suffering from diarrhoea were used as the population of the studies and three studies were conducted on diarrhoeic patients with HIV. More than half of the studies (55.81%) were focusing on

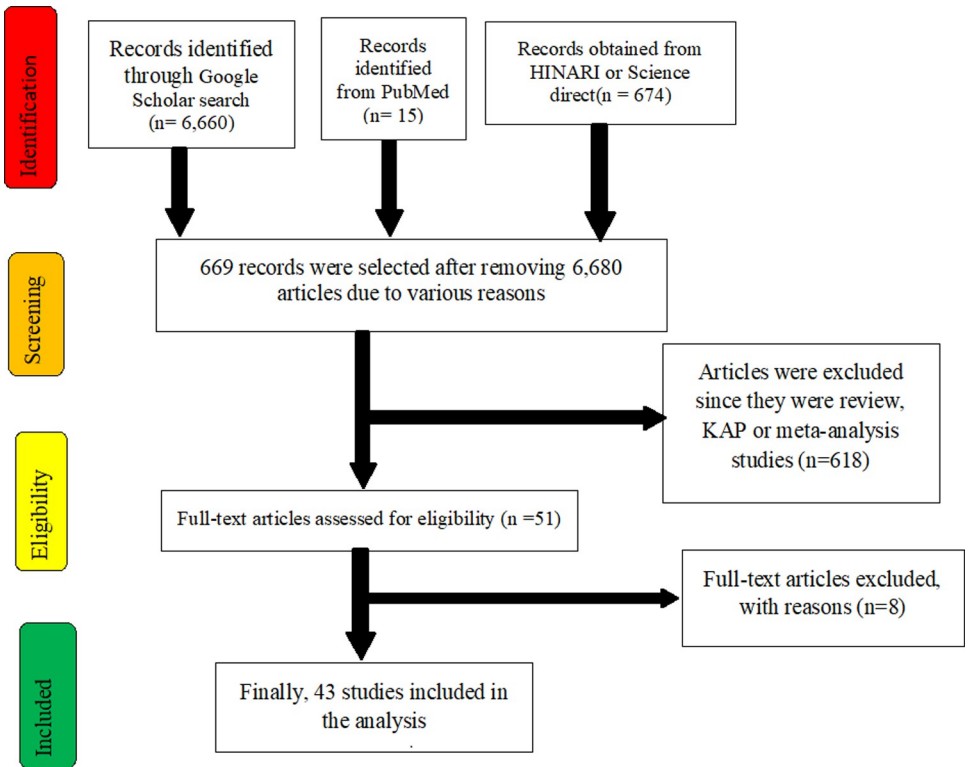

**Fig 1. A flow diagram that shows the selection of articles for the analysis.**

children less than equal to fifteen years of age. However, 39.53% of studies were considering all age groups (Fig 3 and Table 3).

## Prevalence of Gram-negative enteric bacterial pathogens

To isolate GNEBPs, in each study, 24 to 1,225 stool samples were collected. Totally, 13,350 stool samples were examined from 3,688 male and 3,822 female diarrheic patients and 1,962 (14.70%) samples were positive for GNEBPs. The minimum and maximum prevalence of GNEBPs in Ethiopia from diarrhoeic patients were 3.57% [27] and 55.83% [21]. The estimated pooled prevalence of GNEBPs in diarrheic patients from 43 studies was 15.81% (95% CI = 13.33–18.29) (Fig 4).

The distribution of the studies using a funnel plot (Fig 5) showed the asymmetrical distribution of effect estimates; hence, there was a publication bias. To minimize the effect of the bias, subgroup analysis was used. Regionally, the pooled prevalence of GNEBPs from diarrheic patients in Addis Ababa, Amhara, SNNPs and Oromia were 20.08, 16.67, 12.12, and 11.61%, respectively. The pooled prevalence based on the study period was 14. 91, 18.03 and 13.46% among studies from 2006–2010, 2011–2015 and 2016–2021, respectively. The pooled prevalence was the highest (20.35%) in children having age less or equal to 15 years, followed by all age groups (10.83%) and adults greater than 15 years (8.51%) (Table 2).

Table 3 shows the types of bacterial isolates reported by published articles from diarrheic patients. *Shigella* spp. were the most frequent isolate (41.67%), followed by *S. enterica* (38.09%). The pooled estimate of *E. coli* was the highest (19.79%) among enteric bacterial isolates. The pooled prevalence of *Campylobacter* spp., *Shigella* spp. and *Salmonella enterica*. were 10.76, 6.24 and 5.06%, respectively.

Table 1. Characteristics, quality, and the number of Gram-negative isolates recovered from diarrheic patients.

| References | Year of publication | Study period | Region | Study population | Age category | Gender | | Number examined | No. Positive | Prevalence | E. coli | Salmonella | Shigella | Klebsiella | Proteus | Enterobacter | Campylobacter | Citrobacter |
|---|---|---|---|---|---|---|---|---|---|---|---|---|---|---|---|---|---|---|
| | | | | | | Male | Female | | | | | | | | | | | |
| [19] | 2018 | Aug.-Dec 2015 | Addis Ababa | Diarrheic patient | <15 yrs | 115 | 138 | 253 | 94 | 37.15 | 61 | 23 | 10 | | | | | |
| [20] | 2021 | Mar 2019 to Nov 2019 | SNNP | diarrheic patient | Adult > 15yrs | 151 | 127 | 278 | 24 | 8.63 | | 15 | 9 | | | | | |
| [21] | 2018 | Nov 2015 and Aug 2016 | Amhara | diarrheic patient | <15 yrs | 99 | 64 | 163 | 91 | 55.83 | 47 | 5 | 3 | 11 | 7 | 2 | | |
| [22] | 2020 | Jan to March 2018 | Amhara | Diarrheic patient, HIV + | all | 163 | 191 | 354 | 24 | 6.78 | | 17 | 7 | | | | | |
| [23] | 2020 | Jan to July 2014 | Oromia | diarrheic children | <15 yrs | 125 | 114 | 239 | 9 | 3.77 | | 3 | 6 | | | | | |
| [24] | 2018 | June to Sept 2017 | SNNP | diarrheic patient | <15 yrs | 101 | 103 | 204 | 19 | 9.31 | | 3 | 17 | | | | | |
| [25] | 2011 | Aug to Nov 2009 | Amhara | diarrheic patient | all | 125 | 90 | 215 | 32 | 14.88 | | | 32 | | | | | |
| [26] | 2014 | Feb to May, 2014 | Amhara | diarrheic patient | all | 180 | 192 | 372 | 21 | 5.65 | | 4 | 17 | | | | | |
| [27] | 2018 | June to Oct, 2016 | Amhara | diarrheic patient | <15 yrs | 68 | 44 | 112 | 4 | 3.57 | | 1 | 3 | | | | | |
| [28] | 2014 | March to Nov 2012 | Oromia | diarrheic patient | <15 yrs | 114 | 146 | 260 | 22 | 8.46 | | 16 | 6 | | | | | |
| [29] | 2015 | March to May 2011 | Oromia | diarrheic patient | <15 yrs | 14 | 10 | 24 | 7 | 29.17 | | | 7 | | | | | |
| [30] | 2020 | March and Aug 2019 | SNNP | HIV infected diarrheic | Adult >15 yrs | 84 | 96 | 180 | 15 | 8.33 | | 5 | 2 | | | | 8 | |
| [13] | 2011 | Jan to Aug 2006 | Addis Ababa | diarrheic patient | <15 yrs | 654 | 571 | 1225 | 126 | 10.29 | | 65 | 61 | | | | | |
| [31] | 2019 | Nov 2016 and May 2017 | Addis Ababa | diarrheic patient | <15 yrs | 155 | 135 | 290 | 42 | 14.48 | 13 | 7 | 22 | | | | | |
| [32] | 2013 | Oct 2011 to March 2012 | Amhara | diarrheic patient | <15 yrs | 144 | 141 | 285 | 44 | 15.44 | | | | | | | 44 | |
| [33] | 2015 | Dec 2011 to Feb 2012 | Amhara | diarrheic patient | <15 yrs | 239 | 183 | 422 | 73 | 17.30 | | 33 | 40 | | | | | |
| [34] | 2014 | Feb to May 2011 | Harari | diarrheic patient | all | 193 | 191 | 384 | 56 | 14.58 | | | 56 | | | | | |
| [35] | 2014 | June to Oct, 2011 | SNNP | diarrheic patient | <15 yrs | 81 | 77 | 158 | 35 | 22.15 | | 4 | 11 | | | | 20 | |
| [36] | 2014 | Dec 2011 to Feb 2012 | Amhara | diarrheic patient | <15 yrs | | | 422 | 33 | 7.82 | | 33 | | | | | | |
| [37] | 2019 | Feb 2017 to March 2017 | Oromia | diarrheic patient | all | 99 | 133 | 232 | 42 | 18.10 | | 22 | 20 | | | | | |
| [38] | 2020 | Nov 2016 to Jan 2017 | Amhara | diarrheic patient | all | 181 | 203 | 384 | 20 | 5.21 | | 20 | | | | | | |
| [39] | 2015 | Dec 2011 to Feb 2012 | Amhara | diarrheic patient | <15 yrs | 183 | 239 | 422 | 204 | 48.34 | 204 | | | | | | | |
| [40] | 2016 | Dec 2013 to Mar 2014 | Addis Ababa | diarrheic patient | <15 yrs | 125 | 131 | 256 | 78 | 30.47 | 78 | | | | | | | |
| [41] | 2019 | January to March 2018 | Amhara | diarrheic patient | <15 yrs | 151 | 121 | 272 | 29 | 10.66 | | | 29 | | | | | |
| [42] | 2018 | Oct 2015 to Feb 2016 | Oromia | diarrheic patient | all | 223 | 199 | 422 | 39 | 9.24 | | 30 | 9 | | | | | |
| [43] | 2014 | July to Oct 2012 | Oromia | diarrheic patient | <15 yrs | 106 | 121 | 227 | 38 | 16.74 | | | | | | | 38 | |
| [44] | 2018 | Mar to May, 2017 | SNNP | diarrheic patient | <15 yrs | 95 | 72 | 167 | 29 | 17.37 | | 21 | 8 | | | | | |
| [45] | 2019 | June to Dec 2017 | Gambella | diarrheic patient | <15 yrs | 74 | 60 | 134 | 55 | 41.04 | 35 | 4 | 14 | | | | | |
| [46] | 2015 | August to Nov 2014 | Tigray | diarrheic patient | all | 109 | 107 | 216 | 15 | 6.94 | | | 15 | | | | | |
| [47] | 2014 | Oct 2011 to June 2012 | SNNP | diarrheic patient | all | 221 | 161 | 382 | 57 | 14.92 | | 40 | 17 | | | | | |
| [48] | 2011 | Jan to Feb 2007 | Harari | diarrheic patient | all | 119 | 125 | 244 | 45 | 18.44 | | 28 | 17 | | | | | |
| [49] | 2019 | April to July 2016 | Oromia | diarrheic patient | <15 yrs | 179 | 243 | 422 | 47 | 11.14 | | 29 | 18 | | | | | |
| [50] | 2018 | Nov 2011 to March 2012 | Tigray | diarrheic patient | <15 yrs | 145 | 115 | 260 | 37 | 14.23 | | 19 | 18 | | | | | |
| [51] | 2011 | Oct 2006 to March 2007 | Amhara | diarrheic patient | All | 180 | 204 | 384 | 66 | 17.19 | | 60 | 6 | | | | | |
| [52] | 2021 | Apr to Aug 2019 | SNNPR | diarrheic patient | All | 130 | 133 | 263 | 21 | 7.98 | | 1 | 20 | | | | | |
| [53] | 2015 | May 2013 to Jan 2014 | Addis Ababa | Diarrheic patient | All | 425 | 532 | 957 | 59 | 6.17 | | 59 | 4 | | | | | |
| [54] | 2020 | January 2017 | Amhara | Diarrheic patient | <15 yrs | 152 | 192 | 344 | 45 | 13.08 | | 6 | 2 | | | | | |
| [55] | 2016 | - | Oromia | Diarrheic patient | all | 84 | 92 | 176 | 21 | 11.93 | | 19 | | | | | | |
| [56] | 2015 | Aug.-Dec. 2012 | Addis Ababa | Diarrheic patient | <15 yrs | 115 | 138 | 253 | 104 | 41.11 | 61 | 10 | 23 | | | | | 10 |
| [57] | 2017 | Feb to May, 2016 | SNNP | HIV-infected with Gastroenteritis | all | 103 | 112 | 215 | 27 | 12.56 | 2 | 11 | 3 | | | | 13 | |
| [58] | 2018 | Dec. 2014 to March 2015 | Dire Dawa | Diarrheic patient | <15 yrs | 105 | 91 | 196 | 43 | 21.94 | 25 | 7 | 11 | | | | | |
| [59] | 2021 | | Addis Ababa | diarrheic patient | All | 176 | 122 | 298 | 14 | 4.70 | | 14 | | | | | | |
| [60] | 2014 | | Harari | diarrheic patient | All | 208 | 176 | 384 | 56 | 14.58 | | 56 | | | | | | |

yrs = years, SNNP = Southern Nations, Nationalities, and Peoples' Region.

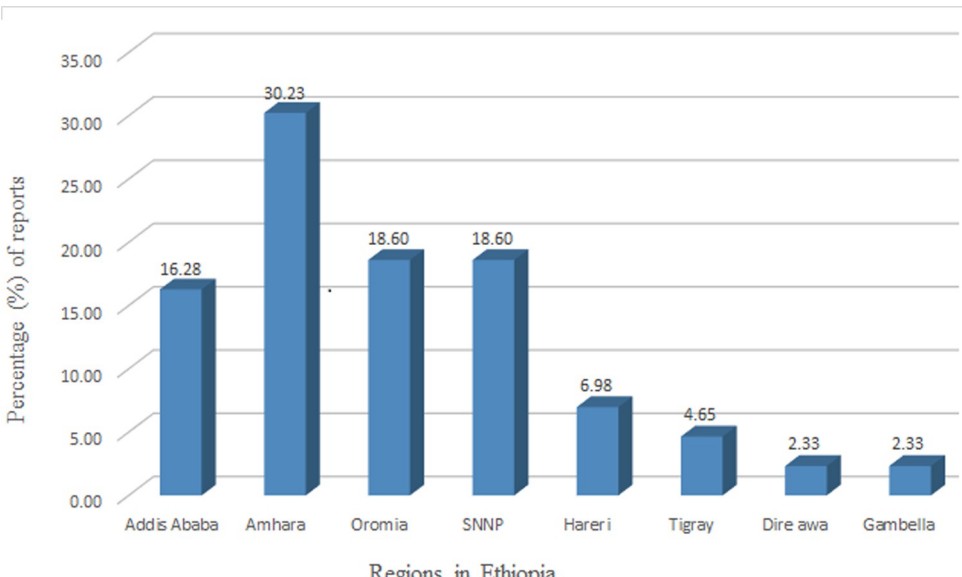

**Fig 2. Percent of published articles in different regions of Ethiopia.** The country has 9 regions and two city administrations. Reports were from 8 of them which are indicated in the figure. Reports were not found from Afar, Somali, and Benishangul Gumuz regions during the period of data collection (SNNP = South Nation and Nationalities Region).

*Escherichia coli* was the most common (15.95%) isolate among children less than or equal to fifteen years of age whereas *Salmonella enterica* and *Shigella* spp. were common among studies that were focused on all age groups (Tables 1 and 4).

## Drug resistance patterns of Gram-negative enteric bacterial pathogens

**Drug resistance in *Shigella* spp.**  Twenty-six antimicrobial panels were used to assess the drug resistance pattern of *Shigella* spp. The highest percentage of resistance was reported among members of the penicillin group such as ampicillin (85.01%) and amoxicillin (82.07%). Resistance against the tetracycline group was also very common (67.01%). A considerable proportion of resistance was also reported among cephalosporin groups, particularly on the first-generation agents and erythromycin. Resistance was not reported against carbapenems (Table 5).

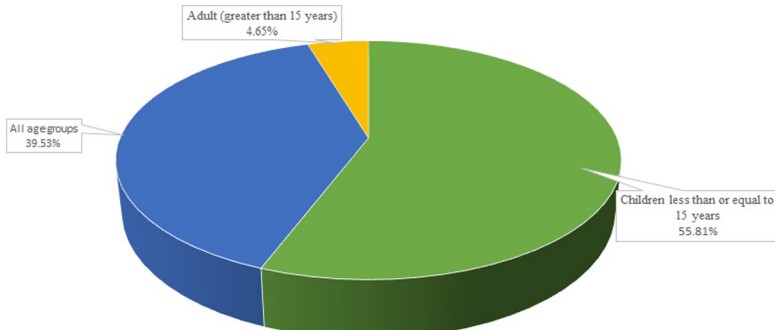

**Fig 3. Age categories and percentage of published articles.**

**Table 2. The pooled prevalence of Gram-negative enterobacterial pathogens from diarrheic patients based on different subgrouping criteria.**

| Subgrouping criteria | Categories | No of studies | Sample examined | No Positive | Pooled prevalence ((%), CI) | I²% (p-value) |
|---|---|---|---|---|---|---|
| Regions | Addis Ababa | 7 | 3532 | 517 | 20.08 (12.85–27.31) | 97.9 (0.00) |
| | Amhara | 13 | 4151 | 686 | 16.67 (10.08–22.48) | 97.5 (0.00) |
| | SNNP | 8 | 1847 | 227 | 12.12 (9.15–15.08) | 75.7 (0.00) |
| | Oromia | 8 | 2002 | 225 | 11.61 (7.98–15.24) | 85.6 (0.00) |
| | Tigray | 2 | 476 | 52 | 10.47 (3.33–17.61) | 85.5 (0.00) |
| | Harari | 3 | 1012 | 157 | 15.39 (13.17–17.67) | 0.00 (0.38) |
| | Dire Dawa | 1 | 196 | 43 | 21.94 (16.15–27.73) | - |
| | Gambella | 1 | 134 | 55 | 41.04 (32.72–49.37) | - |
| Study period | 2006–2010 | 4 | 2068 | 269 | 14.91 (10.44–19.39) | 84.3 (0.00) |
| | 2011–2015 | 21 | 6548 | 1106 | 18.03 (13.70–22.36) | 96.8 (0.00) |
| | 2016–2021 | 18 | 4738 | 587 | 13.46 (10.11–16.80) | 93.8 (0.00) |
| Age of the study subjects | Children <15 years | 24 | 7017 | 1308 | 20.35 (19.92–24.77) | 96.8 (0.00) |
| | Adult > 15 years | 2 | 458 | 39 | 8.51 (5.96–11.07) | 0.00 (0.91) |
| | All age groups | 17 | 5882 | 615 | 10.83 (8.69–12.98) | 88.0 (0.00) |

No = number, % = percent, SNNP = Southern Nations, Nationalities, and Peoples' Region, CI = confident interval, I = Inconsistency Index.

**Drug resistance of *Salmonella enterica*.** Twenty-five antimicrobial panels were used to assess the drug resistance patterns of *S. enterica*. The highest percentage of resistance was reported among members of the penicillin group such as ampicillin (64.98%) and amoxicillin (82.89%). Resistance against tetracycline and trimethoprim-sulphamethoxazole were also

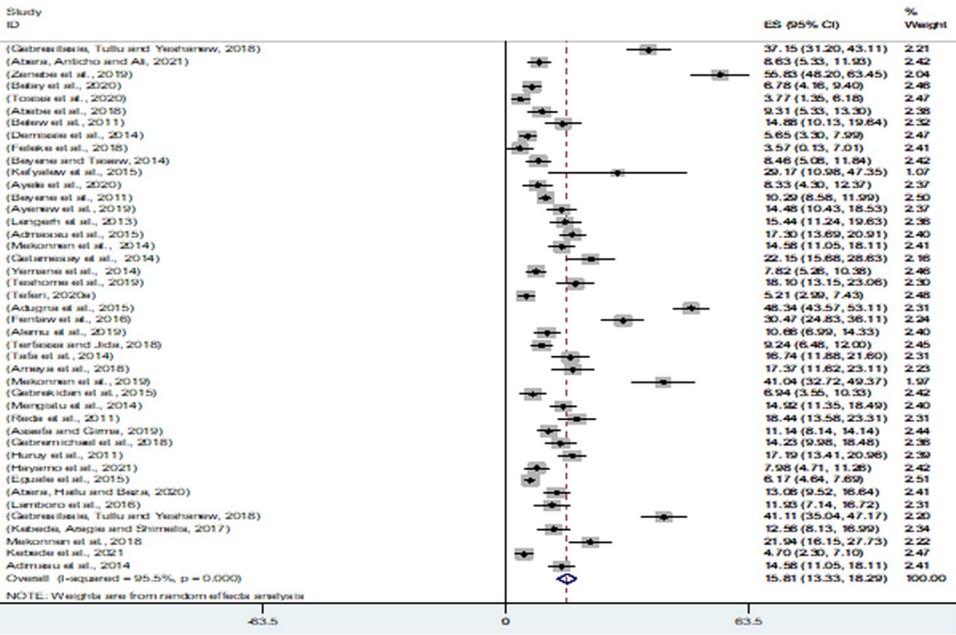

**Fig 4. Forest plot of pooled prevalence estimates of Gram-negative enteric bacterial pathogens among diarrheic patients.** The middle solid vertical line represents the minimum possible prevalence value (0). The dashed line represents the mean pooled prevalence estimate. The black diamond at the centre of the grey box represents the point prevalence estimate of each study and the horizontal line indicates the 95% confidence interval of the estimates. The grey box shows the weight of each study contributing to the pooled prevalence estimate. The last row represents the overall pooled prevalence estimate with a 95% confidence interval.

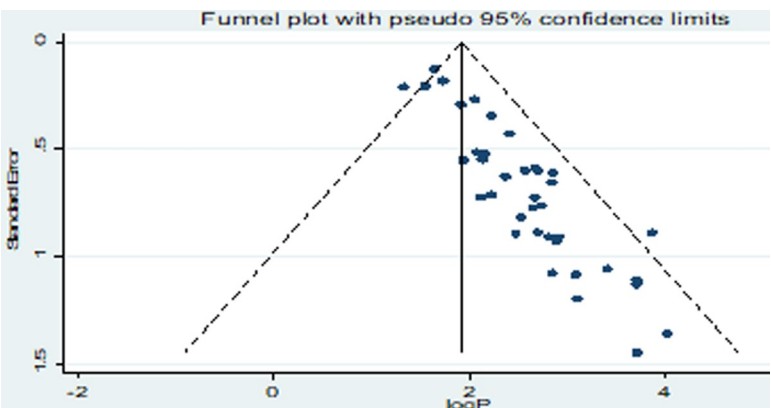

**Fig 5. Funnel plot for the prevalence of Gram-negative enteric bacterial pathogens among diarrheic patients.**

common. A considerable proportion of resistance was also reported against erythromycin and cephalosporin groups. Resistance was not reported against doxycycline, meropenem and azithromycin (Table 6).

**Drug resistance of *Escherichia coli*.** Sixteen antimicrobial panels were used to assess the drug resistance pattern of *E. coli*. The highest percentage of resistance was reported on agents like ampicillin (77.97%). Resistance against tetracycline (76.87%) and trimethoprim-sulphamethoxazole (66.97%) were also common. A considerable proportion of resistance was also reported among cephalosporin groups. Resistance was not reported against meropenem (Table 7).

**Drug resistance of *Campylobacter* spp.** Eighteen antimicrobial panels were used to assess the drug resistance patterns of *Campylobacter* spp. The highest percentage of resistance was

**Table 3. Bacterial isolates reported by published articles from diarrheic patient.**

| Bacterial isolates | No of Studies | Studies reporting the agent (%) | No of sample | No positive | Positives (%) | Prevalence (%) | | |
|---|---|---|---|---|---|---|---|---|
| | | | | | | Minimum | Maximum | Pooled (95% CI) |
| *Shigella* spp. | 35 | 41.67 | 10,026 | 632 | 5.61 | 1.03 | 37.50 | 6.24 (4.66–7.81) |
| *Salmonella enterica* | 35 | 38.09 | 11,360 | 644 | 5.67 | 0.38 | 62.50 | 5.06 (4.04–6.09) |
| *Escherichia coli* | 9 | 7.1 | 2392 | 514 | 21.49 | 0.93 | 51.65 | 19.79 (10.48–29.09) |
| *Campylobacter* spp. | 5 | 6.0 | 1079 | 123 | 11.40 | 4.12 | 16.74 | 10.76 (5.62–15.91) |
| *Citrobacter* spp. | 1 | 1.2 | 253 | 10 | 3.95 | - | - | - |
| *Enterobacter* spp. | 1 | 1.2 | 163 | 2 | 1.23 | - | - | - |
| *Klebsiella* spp. | 1 | 1.2 | 163 | 11 | 6.75 | - | - | - |
| *Proteus* spp. | 1 | 1.2 | 163 | 7 | 4.29 | - | - | - |
| Total | 88* | | | | | | | |

*An article may report one or more bacterial species from diarrheic patients; No = Number; % = Percent; CI = confident interval.

**Table 4. Type bacterial isolates among different age groups.**

| Age category (years) | Type of bacterial Isolates | No of studies | No of Sample | No Positive | Pooled prevalence ((95% CI) |
|---|---|---|---|---|---|
| < 15 | *Campylobacter* spp. | 3 | 670 | 102 | |
| | *E. coli* | 8 | 2177 | 524 | 15.95 (14.52–17.38) |
| | *Salmonella enterica* | 19 | 5893 | 296 | 2.95 (2.52–3.37) |
| | *Shigella* spp. | 20 | 5767 | 344 | 3.95 (3.45–4.44) |
| Adult >15 | *Campylobacter* spp. | 1 | 180 | 8 | |
| | *Salmonella* spp. | 2 | 458 | 20 | |
| | *Shigella* spp. | 2 | 458 | 11 | |
| All age groups | *Campylobacter* spp. | 1 | 215 | 13 | |
| | *E. coli* | 1 | 215 | 2 | |
| | *Salmonella enterica* | 14 | 5067 | 381 | 3.21 (2.74–3.69) |
| | *Shigella* spp. | 13 | 3859 | 221 | 3.14 (2.59–3.68) |

No = Number; % = Percent; CI = confident interval.

**Table 5. Prevalence of *Shigella* spp. resistance to different antimicrobial agents in Ethiopia.**

| Antimicrobial Agents | The Main group of antimicrobial agents | No of studies | The total no of isolates tested | No of resistant isolate | Resistant isolate (%) | Pooled prevalence (%) (95% CI) | I²% (p-value) |
|---|---|---|---|---|---|---|---|
| Ampicillin | Penicillins | 30 | 519 | 452 | 87.09 | 85.01 (79.79–90.22) | 61.4 (0.00) |
| Amoxicillin | Penicillins | 14 | 175 | 153 | 87.43 | 82.07 (74.05–89.65) | 0.00 (0.47) |
| Augmentin (Amoxicillin and clavulanate potassium) | Penicillins and β-lactamase inhibitors | 10 | 196 | 91 | 46.43 | 43.87 (22.5–65.24) | 93.1 (0.00) |
| Tetracycline | Tetracyclines | 22 | 404 | 309 | 76.49 | 67.01 (55.83–78.36) | 89.9 (0.00) |
| Doxycycline | Tetracyclines | 2 | 20 | 16 | 80.00 | - | - |
| Cephalothin | 1st G cephalosporins | 4 | 57 | 40 | 70.18 | 70.18 (60.56–100.65) | 100(-) |
| Cefoxitin | 2nd G cephalosporins | 1 | 20 | 6 | 30.00 | - | - |
| cefuroxime | 2nd G cephalosporins | 1 | 20 | 13 | 65.00 | - | - |
| Cefaclor | 2nd G cephalosporins | 1 | 17 | 11 | 64.71 | - | - |
| Ceftriaxone | 3rd G cephalosporins | 14 | 151 | 16 | 10.60 | 29.53 (7.80–51.25) | 79.5 (0.001) |
| ceftazidime | 3rd G cephalosporins | 6 | 62 | 7 | 11.29 | 8.24 (0.42–16.05) | 0.00 (0.46) |
| cefotaxime | 3rd G cephalosporins | 2 | 20 | 4 | 20.00 | - | - |

(*Continued*)

**Table 5.** (Continued)

| Antimicrobial Agents | The Main group of antimicrobial agents | No of studies | The total no of isolates tested | No of resistant isolate | Resistant isolate (%) | Pooled prevalence (%) (95% CI) | $I^2$% (p-value) |
|---|---|---|---|---|---|---|---|
| Ceftizoxime | 3rd G cephalosporins | 1 | 40 | 11 | 27.5 | - | - |
| Chloramphenicol | Chloramphenicol | 30 | 508 | 220 | 43.31 | 36.95 (29.65–44.24) | 66.7 (0.00) |
| Kanamycin | Aminoglycosides | 2 | 34 | 7 | 20.59 | - | |
| Gentamicin | Aminoglycosides | 27 | 443 | 88 | 19. 86 | 25.38 (17.78–32.99) | 70.4 |
| Amikacin | Aminoglycosides | 3 | 22 | 2 | 9.09 | 14.29 (-4.04–32.62) | 100(-) |
| Streptomycin | Aminoglycosides | 1 | 32 | 31 | 96.87 | - | - |
| Nalidixic acid | Fluoroquinolones | 17 | 191 | 32 | 16.75 | 17.14 (11.34–22.94) | 0.00 (0.69) |
| Norfloxacin | Fluoroquinolones | 14 | 229 | 14 | 6.11 | 8.34 (4.02–12.66) | 0.00 (0.976) |
| Ciprofloxacin | Fluoroquinolones | 28 | 468 | 32 | 6.84 | 11.86 (5.31–18.42) | 65.6 (0.001) |
| Trimethoprim-sulphamethoxazole | Folic acid metabolism inhibitors | 28 | 476 | 270 | 56.72 | 53.00 (44.34–61.67) | 72.90 (0.00) |
| Meropenem | Carbapenems | 2 | 11 | 0 | 0 | - | - |
| Erythromycin | Macrolides | 5 | 38 | 26 | 68.42 | 69.67(46.56–92.77) | 59.9 (0.058) |
| Azithromycin | Macrolides | 2 | 31 | 10 | 32.26 | - | - |
| Clindamycin | Macrolides | 1 | 8 | 4 | 50 | - | - |

No = Number; % = Percent; CI = confident interval, I = Inconsistency Index.

reported for Cephalothin (81.52%). Resistance against ampicillin (65.61%) and trimethoprim-sulphamethoxazole (52.6%) were also common. Resistance was not reported against meropenem and azithromycin (Table 8).

**Drug resistance of other enteric bacterial species.** The number of published articles on other enteric bacterial species from diarrheic patients was very limited and impossible to summarize. However, one study conducted by Zenebe *et al*. [21] reported the presence of *Klebsiella* spp., *Proteus* spp., *Enterobacter* spp. in under-five children with diarrhoea. These bacteria were showing antimicrobial resistance character as indicated in Table 9.

## Multidrug resistance

Out of 43 published articles on enteric bacterial pathogens, 32 (74.42%) reported multidrug-resistant (MDR) characters among the isolates. Among 1470 bacterial isolates, 1104 (75.52%) with a pooled prevalence of 70.56% (CI = 64.56–76.77%) were resistant to three or more antimicrobial agents (multidrug resistance). The report showed that the pooled prevalence of MDR in *Campylobacter* spp., *Shigella* spp., *E. coli* and *S. enterica* were 80.78, 79.08, 78.20 and 59.46%, respectively (Table 10).

**Table 6. The pooled prevalence of *Salmonella enterica* resistance to different antimicrobial agents in Ethiopia.**

| Antimicrobial Agents | The main group of antimicrobial agents | No of studies | The Total no of isolates tested | No of resistant isolate | Resistant isolate (%) | Pooled prevalence (%) (95% CI) | I²% (p-value) |
|---|---|---|---|---|---|---|---|
| Ampicillin | Penicillins | 29 | 588 | 447 | 76.02 | 64.98 (45.2–84.76) | 96.5 (0.00) |
| Amoxicillin | Penicillins | 12 | 255 | 223 | 87.45 | 82.89 (70.58–95.21) | 62.0 (0.072) |
| Augmentin (Amoxicillin and clavulanate potassium) | Penicillins and β-lactamase inhibitors | 8 | 185 | 75 | 40.54 | 45.34 (19.11–71.56) | 95.2 (0.00) |
| Tetracycline | Tetracycline | 24 | 555 | 287 | 51.71 | 54.59 (41.28–67.90) | 90.03 (0.00) |
| Doxycycline | Tetracycline | 2 | 34 | 0 | 0.00 | - | - |
| Cephalothin | 1st G cephalosporins | 2 | 9 | 1 | 11.11 | - | - |
| Cefaclor | 2nd G cephalosporins | 1 | 4 | 4 | 100.00 | - | - |
| Cefoxitin | 2nd G cephalosporins | 1 | 33 | 9 | 27.27 | - | - |
| Ceftizoxime | 2nd G cephalosporins | 1 | 21 | 5 | 23.81 | - | - |
| cefuroxime | 2nd G cephalosporins | 2 | 55 | 16 | 29.09 | - | - |
| Ceftriaxone | 3rd G cephalosporins | 16 | 384 | 109 | 28.39 | 33.1 (7.88–58.33) | 98.0 (0.00) |
| ceftazidime | 3rd G cephalosporins | 5 | 27 | 4 | 14.81 | 29.22 (22.42–8086) | 81.7 (0.02) |
| cefotaxime | 3rd G cephalosporins | 2 | 7 | 2 | 28.57 | - | - |
| Chloramphenicol | Chloramphenicol | 29 | 629 | 260 | 41.34 | 42.39 (27.39–57.38) | 96.1 (0.00) |
| Kanamycin | Aminoglycosides | 3 | 73 | 24 | 32.88 | 33.7 (22.72–44.69) | 0.00 (0.67) |
| Gentamicin | Aminoglycosides | 24 | 491 | 121 | 24.64 | 17.36 (7.57–27.15) | 93.4 (0.00) |
| Amikacin | Aminoglycosides | 3 | 26 | 1 | 3.85 | - | - |
| Nalidixic acid | Fluoroquinolones | 19 | 462 | 66 | 14.29 | 14.60 (9.23–19.97) | 64.2 (0.00) |
| Norfloxacin | Fluoroquinolones | 10 | 200 | 9 | 4.50 | 5.17 (1.79–8.55) | 0.00 (0.98) |
| Ciprofloxacin | Fluoroquinolones | 28 | 28 | 28 | 5.29 | 7.66 (2.67–12.66) | 74.8 (0.00) |
| Trimethoprim-sulphamethoxazole | Folic acid metabolism inhibitors | 25 | 464 | 218 | 46.98 | 46.72 (29.74–61.69) | 94.4 (0.00) |
| Meropenem | Carbapenems | 2 | 20 | 0 | 0.00 | - | - |
| Erythromycin | Macrolides | 5 | 45 | 25 | 55.56 | 52.97 (37.96–68.72) | 5.8 (0.364) |
| Azithromycin | Macrolides | 1 | 5 | 0 | 0.00 | - | - |

(*Continued*)

**Table 6.** (Continued)

| Antimicrobial Agents | The main group of antimicrobial agents | No of studies | The Total no of isolates tested | No of resistant isolate | Resistant isolate (%) | Pooled prevalence (%) (95% CI) | I²% (p-value) |
|---|---|---|---|---|---|---|---|
| Clindamycin | Macrolides | 1 | 113 | 1 | 0.88 | - | - |

No = Number; % = Percent; CI = confident interval, I = Inconsistency Index.

## Discussion

Diarrhoea is a common health problem, causing mortality and morbidity for thousands of people around the globe. Both infectious and non-infectious agents can induce the problem, among the infectious agents, enteric bacterial pathogens like diarrheagenic *E. coli*, *S. enterica*, *Shigella* spp., and *Campylobacter* spp. play important roles in the induction or severity of diarrhoea [61]. Determining their burden in a given population is very essential to design strategies

**Table 7. The pooled prevalence of *Escherichia coli* resistance to different antimicrobial agents in Ethiopia.**

| Antimicrobial Agents | The main group of antimicrobial agents | No of studies | The total no of isolates tested | No of resistant isolate | Resistant isolate (%) | Pooled prevalence (%) (95% CI) | I² (p-value) |
|---|---|---|---|---|---|---|---|
| Ampicillin | Penicillins | 6 | 446 | 359 | 80.49 | 77.97 (70.17–85.76) | 71.4 (0.00) |
| Amoxicillin | Penicillins | 1 | 47 | 5 | 10.64 | - | - |
| Augmentin (Amoxicillin and clavulanate potassium) | Penicillins and B-lactamase inhibitors | 5 | 339 | 204 | 60.18 | 64.78 (42.00–87.57) | 95.0 (0.00) |
| Tetracycline | Tetracyclines | 3 | 253 | 194 | 76.68 | 76.87 (71.69–82.05) | 0.00 (0.563) |
| Ceftriaxone | 3ʳᵈ G cephalosporins | 5 | 284 | 11 | 3.87 | 2.91 (0.74–5.08) | 11.5 (0.34) |
| Cephalothin | 1st G cephalosporins | 1 | 47 | 8 | 17.02 | - | - |
| cefotaxime | 3ʳᵈ G cephalosporins | 1 | 204 | 50 | 24.51 | - | - |
| Gentamicin | Aminoglycosides | 6 | 388 | 102 | 26.29 | 19.72 (7.41–32.03) | 88.03 (0.00) |
| Amikacin | Aminoglycosides | 1 | 113 | 2 | 1.77 | - | - |
| Chloramphenicol | Chloramphenicol | 5 | 375 | 121 | 32.27 | 30.39 (20.95–39.84) | 67.0 (0.016) |
| Nalidixic acid | Fluoroquinolones | 5 | 224 | 38 | 16.96 | 16.71 (11.81–21.6) | 0.00 (0.713) |
| Norfloxacin | Fluoroquinolones | 2 | 206 | 20 | 9.71 | - | - |
| Ciprofloxacin | Fluoroquinolones | 7 | 439 | 27 | 6.15 | 5.57 (3.42–7.71) | 0.00 (0.063) |
| Trimethoprim-sulphamethoxazole | Folic acid metabolism inhibitors | 7 | 428 | 301 | 70.33 | 66.97 (56.21–77.71) | 80.7 (0.00) |
| Meropenem | Carbapenems | 1 | 133 | 0 | 0 | - | - |
| Erythromycin | Macrolides | 1 | 2 | 1 | 50 | - | - |

No = Number; % = Percent; CI = confident interval; I = Inconsistency Index.

**Table 8. Pooled prevalence of *Campylobacter* spp. resistance to different antimicrobial agents in Ethiopia.**

| Antimicrobial Agents | The main group of antimicrobial agents | No of studies | The total no of isolates tested | No of resistant isolate | Resistant isolate (%) | Pooled prevalence (%) (95% CI) | I2 (p-value) |
|---|---|---|---|---|---|---|---|
| Ampicillin | Penicillins | 4 | 110 | 72 | 65.45 | 65.61(44.90–86.32) | 83.3 (0.000 |
| Amoxicillin | Penicillins | 1 | 20 | 16 | 80.00 | - | - |
| Augmentin (Amoxicillin and clavulanate potassium) | Penicillins and B-lactamase inhibitors | 1 | 44 | 16 | 36.36 | - | - |
| Tetracycline | Tetracyclines | 5 | 123 | 60 | 48.78 | 42.17 (20.30–64.05) | 85.4 (0.00) |
| Doxycycline | Tetracyclines | 3 | 90 | 16 | 17.78 | 18.94 (10.5–27.38) | 0.00 (0.379) |
| Ceftriaxone | 3$^{rd}$ G cephalosporins | 4 | 85 | 15 | 17.65 | 39.43 (0.96–77.91) | 79.0 (0.029) |
| ceftazidime | 3$^{rd}$ G cephalosporins | 1 | 8 | 2 | 25.00 | - | - |
| Cephalothin | 1$^{st}$ G cephalosporins | 3 | 102 | 91 | 89.22 | 81.52 (63.78–99.27) | 63.2 (0.09) |
| Gentamycin | Aminoglycosides | 5 | 123 | 29 | 23.58 | 30.70 (8.04–53.36) | 88.0 (0.00) |
| Chloramphenicol | Chloramphenicol | 5 | 123 | 26 | 21.14 | 28.03 (10.33–45.85) | 74.4 (0.00) |
| Nalidixic acid | Fluoroquinolones | 4 | 115 | 13 | 11.30 | 10.45 (4.89–16.00) | 0.00 (0.711) |
| Norfloxacin | Fluoroquinolones | 3 | 95 | 10 | 10.53 | 12.02 (4.99–19.05) | 0.00 (0.665) |
| Ciprofloxacin | Fluoroquinolones | 5 | 123 | 19 | 15.45 | 13.9 (7.87–19.94) | 0.00 (0.528) |
| Trimethoprim-sulphamethoxazole | Folic acid metabolism inhibitors | 5 | 123 | 67 | 54.47 | 52.6 (33.76–71.43) | 78.4 (0.00) |
| Meropenem | Carbapenems | 1 | 8 | 0 | 0.00 | - | - |
| Erythromycin | Macrolides | 5 | 123 | 37 | 30.08 | 35.72 (18.34–35.10) | 78.4 (0.001) |
| Azithromycin | Macrolides | 1 | 8 | 0 | 0.00 | - | - |
| Clindamycin | Macrolides | 2 | 82 | 28 | 34.15 | - | - |

No = Number; % = Percent; CI = confident interval; I = Inconsistency Index.

for the reduction of the incidence and influences of diarrhoea. The minimum and maximum prevalence of GNEBPs in Ethiopia from diarrhoeic patients were 3.57% [27] and 55.83% [21], respectively. The pooled prevalence of EBP isolates from the stool of diarrheic patients in Ethiopia was 15.81% (CI = 13.33–18.29). In line with this finding, Getie *et al.* [62] reported 13.2% prevalence of GNEBP in other groups of population (food handlers) in Gondar town, Northwest Ethiopia. On the other hand, Shah *et al.* [63] reported a 33.62% prevalence of GNEBP in Kenya. The difference may be due to the detection methods since they use molecular techniques in addition to the conventional culturing methods.

Almost all studies (97.67%) were institutional, and samples were collected from patients visiting health facilities. Focusing only on health facilities may not reflect the overall prevalence of GNEBPs and their drug resistance patterns in the country. Since some GNEBP infection cases

**Table 9. Pooled prevalence of other enteric bacterial resistance to different antimicrobial agents in Ethiopia.**

| References | bacterial isolate | Ampicillin | | | Chloramphenicol | | | Gentamicin | | | Nalidixic acid | | | Tetracycline | | | Ciprofloxacin | | | Trimethoprim-sulphamethoxazole | | | Ceftriaxone | | | Amoxicillin | | | Cephalothin | | |
|---|---|---|---|---|---|---|---|---|---|---|---|---|---|---|---|---|---|---|---|---|---|---|---|---|---|---|---|---|---|---|---|
| | | No isolate tested | No resistance | % | No isolate tested | No resistance | % | No isolate tested | No resistance | % | No isolate tested | No resistance | % | No isolate tested | No resistance | % | No isolate tested | No resistance | % | No isolate tested | No resistance | % | No isolate tested | No resistance | % | No isolate tested | No resistance | % | No isolate tested | No resistance | % |
| [21] | *Klebsiella* | 11 | 3 | 27.27 | 11 | 2 | 18.18 | 11 | 4 | 36.36 | 11 | 1 | 9.09 | 11 | 9 | 81.82 | 11 | 0 | 0.00 | 11 | 8 | 72.73 | 11 | 0 | 0.00 | 11 | 4 | 36.36 | 11 | 2 | 18.18 |
| [21] | *Proteus* | 7 | 2 | 28.57 | 7 | 1 | 14.29 | 7 | 3 | 42.86 | 7 | 4 | 57.14 | 7 | 2 | 28.57 | 7 | 0 | 0.00 | 7 | 2 | 28.57 | 7 | 1 | 14.29 | 7 | 3 | 42.86 | 7 | 0 | 0.00 |
| [21] | *Enterobacter* | 2 | 1 | 50.0 | 2 | 0 | 0.00 | 2 | 1 | 50.0 | 2 | 1 | 50.0 | 2 | 0 | 0.0 | 2 | 0 | 0.0 | 2 | 0 | 0.0 | 2 | 0 | 0.0 | 2 | 0 | 0.0 | 2 | 0 | 0.00 |

**Table 10. Multidrug resistance pattern of enteric bacteria pathogens from diarrheic patients.**

| Type bacterial isolate | No of studies | The total no of isolates tested | No of multidrug-resistant isolates | Multidrug-resistant isolate (%) | Pooled prevalence (%) (95% CI) | I² (p-value) |
|---|---|---|---|---|---|---|
| *Shigella* spp. | 24 | 443 | 363 | 81.94 | 79.08 (72.19–85.97) | 68.2 (0.00) |
| *Salmonella enterica* | 23 | 496 | 317 | 63.91 | 59.46 (46.13–72.79) | 91.1 (0.00) |
| *Escherichia coli* | 6 | 410 | 332 | 80.98 | 78.20 (67.46–88.93) | 84.7 (0.00) |
| *Campylobacter* spp. | 4 | 110 | 87 | 79.09 | 80.78 (65.04–96.52) | 94.6 (0.00) |
| *Klebsiella* spp. | 1 | 11 | 5 | 45.45 | - | - |
| | | 1470 | 1104 | 75.52 | | |

No = number, % = percent; CI = confidence interval, I = Inconsistency Index.

may not arrive at health institutions and widespread use or misuse of antimicrobial drugs in the community may accelerate the occurrence of antimicrobial resistance [64,65].

Sub-grouping of the prevalence of GNEBPs based on the studies conducted in different regions of the country showed that the pooled prevalence was high in Addis Ababa (20.08%). In line with this, a spatial variation across the regions of Ethiopia was reported by Bogale *et al*. [66]. The difference based on the study period was not significant. However, a declining pattern of diarrhoea at the national level was reported by Bogale *et al*. [66]. Published articles were from 8 regions of Ethiopia, published articles were not found in Afar, Somali and Benishangul Gumuz regions of the country in this study. Hence, the pooled prevalence was calculated from 8 regions ignoring others. However, the scenario may not be equivalent to the region of the country having no reports.

Grouping of study participants based on their age showed that the pooled prevalence was the highest (20.35%) in children having age below or equal to 15 years which indicated that children are more exposed to the GNEBPs and express severe syndromes to visit health facilities. In line with this, Kotloff [4] and Havelaar *et al*. [67] reported that children contribute a huge proportion of diarrheal diseases in the world.

Diarrheagenic *E. coli*, *Shigella* spp., *S. enterica* and *Campylobacter* spp. were the most common isolates among GNEBPs. In line with this, Getie *et al*. [62] reported that *Shigella* spp. enterohemorrhagic *E. coli* (EHEC) and *S. enterica* were important isolates of GNEBPs among food handlers. In this study, the pooled estimate of *E. coli* was the highest (19.79%) among enteric bacterial pathogens. The result is very close to the report of Zenebe *et al*. [68] who reported that the pooled prevalence of *E. coli* was 25% in Ethiopia. A 33.8% pooled prevalence was reported by Oppong *et al*. [69] in Sub-Sharan Africa. Oppong *et al*. [69] also found that *E. coli* detection was the highest in the East African region and lowest in the middle part of Africa. The difference may be due to the number of studies in the summary and the targeted strain of *E. coli*. For example, in a single study in Niger, 11.1% of diarrhoeic children were positive for diarrheagenic *E. coli* [70]. Similarly, the Global enteric multicentre study on infants and children showed that diarrheagenic *E. coli* was among the four major pathogens responsible for diarrhoea in low-income and middle-income countries [71].

The pooled prevalence of *Campylobacter* spp. in this analysis was 10.76%. In line with this report, Kassie *et al*. [72] reported a prevalence of 10.5% among children in Denbia district, Ethiopia and 13.8% prevalence was also reported by Gedlu and Aseffa [73] among children in northwest Ethiopia. Oppong *et al*. [69] reported a 12.3% pooled prevalence of *Campylobacter*

in East Africa. A high rate of *Campylobacter* infection among children was also reported in Kenya [63]. In contrary to the report of this study, Fletcher *et al.* [74] reported a pooled prevalence of 2.7% in Sub-Saharan countries. The difference may be related to the area coverage, disease prevention and control practices.

The pooled prevalence of *Shigella* spp. in this study was 6.24% which is equivalent to the report of Hussen *et al.* [75]. According to their report, the pooled prevalence of *Shigella* spp. in Ethiopia was 6.6%. Similarly, Oppong *et al.* [69] reported a pooled prevalence of 5.6% from children under five and Fletcher *et al.* [74] 4.3% of children aged less than 12 years in Sub-Saharan countries.

The pooled prevalence of *S. enterica* was 5.06% which is almost comparable with the reported pooled prevalence by Abate and Assefa [76], which was 4.8% among human stools and animal origin foods in Ethiopia.

Resistance to antimicrobial agents is a natural evolutionary process for the bacteria, however, the process is accelerated by human activities in terms of antimicrobial usage patterns and infection control or prevention practices. The risks are very high in developing countries like Ethiopia where there is a widespread use or misuse of antimicrobial agents with a high burden of infectious diseases [76–79].

In this meta-analysis, *Shigella* isolates were more resistant to the penicillin group of antimicrobial agents like ampicillin (85.01%) and amoxicillin (82.07%). A high percentage of resistance against ampicillin (83.1%) and amoxicillin (84.1%) were also reported by Hussen *et al.* [75]. There were also reports of drug resistance among the first-line drugs like ciprofloxacin (11.86%) and ceftriaxone (29.53%) for the treatment of Shigellosis. Resistance development against such types of antimicrobial agents was also reported by Hussen *et al.* [75]. They reported 8.9 and 9.3% resistance against ciprofloxacin and ceftriaxone, respectively.

In this analysis, high proportions of *Salmonella* isolates were also resistant against the penicillin group of antimicrobial agents like ampicillin (64.98%) and amoxicillin (82.89%). In line with this Tadesse [80] was also reported a high percentage (86.01%) of resistance of *Salmonella* against ampicillin and/or amoxicillin. Resistance of *Salmonella* isolates to fluoroquinolone like ciprofloxacin (2.27%) and third-generation cephalosporin (ceftriaxone 16.68%) were also reported. Similarly, according to Tadesse's report, the pooled prevalence of ciprofloxacin resistance among *Salmonella* isolates was 3.61% [80] and a prevalence of 2.9% of resistance against ciprofloxacin was also reported in Iran [81].

Resistance was common among *E. coli* isolates in this analysis, particularly on antimicrobial agents like ampicillin (77.97%), tetracycline (76.87%) and trimethoprim-sulphamethoxazole (66.97%). Resistance of *E. coli* against a wide array of antimicrobials was also reported by Pormohammad *et al.* [82], Zenebe *et al.* [68] and Tuem *et al.* [83]. Resistance in non-pathogenic strains of *E. coli* may not have a direct effect on health, however, non-pathogenic resistant strains may acquire virulence genes and induce disease that may not be treated easily or non-pathogenic strains having resistant character may act as a reserve for the resistant character for other bacteria [82].

Among *Campylobacter* isolates in this analysis, the highest percentage of resistance was reported on antimicrobial agents like cephalothin (81.52%), ampicillin (65.61%) and trimethoprim-sulphamethoxazole (52.60%). A resistance pattern of 92.3–100% to erythromycin and the β—lactams, 61.5–86.7% to trimethoprim-sulfamethoxazole, 92.3–93.3% to tetracycline, 46.2–80% to chloramphenicol, 0–60% to aminoglycosides and 0% to imipenem were reported among *Campylobacter* spp. in Ghana [84].

In this study, among 1470 bacterial isolates 1104 (75.52%) with a pooled prevalence of 70.56%, were resistant to three or more antimicrobial agents (multidrug resistance) (MDR). In line with this finding, Alemayehu [85] reported that the pooled prevalence of multidrug resistance was

70.5% among bacterial isolates in Ethiopia. Another, meta-analysis study on multidrug resistance by Abayneh et al. [86] reported the pooled prevalence of 80.5% among Gram-negative bacteria. It was also very close to the reports from India (66.12%) [87] and Egypt (65.5%) [88,89]. In contrast to our finding in Ethiopia, a lower prevalence of MDR has been reported from Germany (60%) [90], Nepal (42.6%) [91], Australia (36%) [92], Indonesia (28.7%) [93], the USA (27%) [94], Spain (34.5%) [95] and France (11.6%) [96]. Several factors may play a role in the difference including the magnitude and style of antimicrobial use and infection prevention practices.

The MDR report among *Campylobacter* isolates in this analysis (80.78%) was higher than the report from Bangladesh (28.8%) [97] and Kenya 50% [98] but lower than the report from Ghana (97%) [84]. This may be due to differences in the use of antimicrobial agents, the area coverage, sample type and technique of detections. The MDR report of *E. coli* in this analysis (78.20) was not in line with other reports like 50% prevalence in Nigeria [99], 40% in Spain [95], 22% among human isolates in the world [82], 26% in China [100], 39.8% in Egypt isolates from animals [101], 28% in low and middle-income countries [102]. The MDR report of this meta-analysis among *Shigella* isolates (79.08) was almost similar to the report by Hussen et al. [75] (83.2%) but it was lower than other reports from Iran (89.4%) [103], and Bangladesh (94%) [104]. However, it was higher than other reports such as 53.8% among migrants in Europe [74], 60% in Kenya [98] and 19% in Somalia [105]. Factors like the magnitude and style of antimicrobial use and infection prevention practices may play roles in the differences.

The development of MDR character among *S. enterica* is also an important public health concern around the globe [106]. In this analysis, the prevalence of MDR character among *Salmonella* isolate was 59.46% which is comparable with the MDR report of Garedew et al. [107] (46.2%) among food handlers in Gondar. However, it was lower than the report from Dagnew et al. [108] (76%) and Admassu et al. [109] (100%). The difference may be due to the target population and study methods (single versus pooled meta-analysis reports).

## Limitations of the study

Some of the studies included in the analysis were targeting the most common bacterial pathogens that did not rule out the absence of others. Therefore, for bacteria that are thought to be less frequent, the reported prevalence may not accurate. Publication bias and heterogenicity were observed in the analysis, but attempts were made to reduce their impact on the analysis by following the random effect model and subgrouping. However, these may not totally avoid their impact on the interpretation of the pooled results.

## Conclusion

According to this analysis, the burden of Gram-negative enteric bacterial pathogens (GNEBPs) was high and may be considered a major cause of diarrhoea in Ethiopia. A significant proportion of the isolates exhibited resistance to the commonly used antibacterial agents which are expected to affect the treatment response and cost, morbidity and progression of the infection. *Shigella* spp., *S. enterica*, *E. coli* and *Campylobacter* spp. were the commonly isolated GNEPB from diarrheic patients in the country. The pooled estimate of *Campylobacter* spp. was the highest followed by *E. coli*, *Shigella* spp. and *S. enterica*. Resistance to antimicrobial agents was most common among the penicillin groups, followed by tetracycline, and trimethoprim-sulphamethoxazole. However, there were also resistant strains against very relevant drugs for the treatment of GNEBP such as fluoroquinolones and thirdgeneration cephalosporins. Almost all isolates were susceptible to meropenem.

Since the incidences of these bacterial diseases are related to hygiene, all activities that enhance hygienic practices (clean water and food, handwashing, proper use of latrine) must be

advocated and implemented. Performing drug sensitivity tests for suspected diarrheagenic bacteria is extremely advantageous to select the appropriate antimicrobial drugs for the treatment. The antimicrobial resistance surveillance system must be established to understand the trend of resistance among pathogenic bacteria and to plan and implement mitigating strategies like proper control and prevention of infectious diseases and antimicrobial stewardship programs. Almost all studies were using conventional techniques for confirmation of the isolates, thus, adding molecular methods in the future will increase analysis precision.

## Supporting information

**S1 File. Forest plot of pooled prevalence estimates of Gram-negative enteric bacterial pathogens in different regions of Ethiopia.**
(DOCX)

**S2 File. Forest plot of pooled prevalence estimates of Gram-negative enteric bacterial pathogens subgrouping based on the study period.**
(DOCX)

**S3 File. Forest plot of pooled prevalence estimates of Gram-negative enteric bacterial pathogens subgrouping-based age of the study subjects.**
(DOCX)

**S4 File. Frost plot of the studies on *Shigella* species.**
(DOCX)

**S5 File. Frost plot of the studies on *Salmonella enterica*.**
(DOCX)

**S6 File. Frost plot of the studies on *Escherichia coli* strains.**
(DOCX)

**S7 File. Frost plot of the studies on *Campylobacter* species.**
(DOCX)

**S8 File. Frost plot of the studies on multidrug resistance.**
(DOCX)

**S9 File. Frost plot of the studies on multidrug resistance of *Shigella* species.**
(DOCX)

**S10 File. Frost plot of the studies on multidrug resistance of *Salmonella enterica*.**
(DOCX)

**S11 File. Frost plot of the studies on multidrug resistance of *E. coli*.**
(DOCX)

**S12 File. Frost plot of the studies on multidrug resistance of *Campylobacter* species.**
(DOCX)

**S13 File. Frost plot of the studies on the prevalence of *Escherichia coli* in children less than or equal to 15 years of age.**
(DOCX)

**S14 File. Frost plot of the studies on the prevalence of *Escherichia coli* in children less than or equal to 15 years of age.**
(DOCX)

**S15 File. Frost plot of the studies on the prevalence of *Shigella* spp in children less than or equal to 15 years of age.**
(DOCX)

**S16 File. Frost plot of the studies on the prevalence of *Salmonella enterica* in all age groups.**
(DOCX)

**S17 File. Frost plot of the studies on the prevalence of *Shigella* species in all age groups.**
(DOCX)

## Acknowledgments

The authors would like to thank the researchers who publish their research work and make it available to the public through the internet.

## Author Contributions

**Conceptualization:** Achenef Melaku Beyene, Mucheye Gezachew, Ahmed Yousef, Baye Gelaw.

**Data curation:** Achenef Melaku Beyene.

**Formal analysis:** Achenef Melaku Beyene.

**Methodology:** Achenef Melaku Beyene, Desalegn Mengesha, Ahmed Yousef, Baye Gelaw.

**Software:** Achenef Melaku Beyene.

**Supervision:** Desalegn Mengesha, Ahmed Yousef, Baye Gelaw.

**Validation:** Achenef Melaku Beyene, Baye Gelaw.

**Writing – original draft:** Achenef Melaku Beyene.

**Writing – review & editing:** Achenef Melaku Beyene, Mucheye Gezachew, Desalegn Mengesha, Ahmed Yousef, Baye Gelaw.

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
