## [Decision Letter · Decision Letter 0]

17 Jan 2022

PONE-D-21-39208Prevalence and drug resistance patterns of Gram-negative enteric bacterial pathogens from diarrheic patients in Ethiopia: A systematic review and meta-analysisPLOS ONE

Dear Dr. Beyene,

Thank you for submitting your manuscript to PLOS ONE. After careful consideration, we feel that it has merit but does not fully meet PLOS ONE’s publication criteria as it currently stands. Therefore, we invite you to submit a revised version of the manuscript that addresses the points raised during the review process.

Many thanks for submitting your manuscript to PLOS One

It was reviewed by two experts in the field, and they have recommended some minor modifications be made prior to acceptance

I therefore invite you to make these changes and to write a response to reviewers which will expedite revision upon resubmission

I wish you the best of luck with your modifications

Hope you are keeping safe and well in these difficult times

Thanks

Simon==============================

We look forward to receiving your revised manuscript.

Kind regards,

Simon Clegg, PhD

Academic Editor

PLOS ONE

Journal Requirements:

4. We note that Figure 2 in your submission contain map images which may be copyrighted. All PLOS content is published under the Creative Commons Attribution License (CC BY 4.0), which means that the manuscript, images, and Supporting Information files will be freely available online, and any third party is permitted to access, download, copy, distribute, and use these materials in any way, even commercially, with proper attribution. For these reasons, we cannot publish previously copyrighted maps or satellite images created using proprietary data, such as Google software (Google Maps, Street View, and Earth). For more information, see our copyright guidelines: http://journals.plos.org/plosone/s/licenses-and-copyright.

a) You may seek permission from the original copyright holder of Figure 2 to publish the content specifically under the CC BY 4.0 license.  

5. We note you have included a table to which you do not refer in the text of your manuscript. Please ensure that you refer to Table 1 in your text; if accepted, production will need this reference to link the reader to the Table.

Reviewers' comments:

Reviewer's Responses to Questions

**Comments to the Author**

1. Is the manuscript technically sound, and do the data support the conclusions?

Reviewer #1: Yes

Reviewer #2: Yes

2. Has the statistical analysis been performed appropriately and rigorously? 

Reviewer #1: Yes

Reviewer #2: Yes

3. Have the authors made all data underlying the findings in their manuscript fully available?

Reviewer #1: Yes

Reviewer #2: Yes

4. Is the manuscript presented in an intelligible fashion and written in standard English?

Reviewer #1: Yes

Reviewer #2: Yes

5. Review Comments to the Author

Reviewer #1: 1.Published articles were not found in Afar, Somali, and Benishangul Gumuz regions of the country in this study.It is suggested to be analysed in the discussion section.

2. Table 1 lists the detection techniques and methods used in the article should be better.

3. Analysis of Common Pathogens by Adults and Children in table 3 maybe useful.

4. From table 5 to table 7, If there are too few isolates for statistical analysis, it is recommended to delete relevant information to avoid bias in drug resistance

5.Analysis of Multi-drug resistance pattern by Adults and Children in table 9 maybe useful.

Reviewer #2: The manuscript attempts to synthesize information on facility based prevalence of GNEBPs in Ethiopia and determine their AMR patterns. It is extremely important to understand the extent of GNEBP infections and the current resistance profiles for not only instituting surveillance and intervention but also for choosing the right antibiotics for emperical treatment whenever required. Therefore the study is appropriate and in context. It has been done by employing standard PRISMA guidelines.

I have the following minor comments that I feel should be addressed by the authors at appropriate sections in the manuscript.

1. The study primarily focuses on E.coli, Campylobacter, Shigella, and Salmonella. While all except E.coli are known to be pathogenic, there is no description on how the studies differentiated DEC from commensal E.coli. The authors have touched upon the fact that resistance of commensal E.coli may not be relevant but may affect those of DEC but methods employed by the various reports to identify DEC is completely missing. In particular, it is said that the studies used conventional methods and not molecular methods and therefore it leaves us wondering how they were differentiated.

2. Was there any information available on co-infections?

3. We often see that GNEBPs constitute only a minor portion of childhood diarrhoeas. For eg. Rotavirus is responsible for almost 30-40% of diarrhoeas in developing countries. In one particular study in India, viral infections accounted for almost 80% of all cases of diarrhoea. Therefore the rampant use/misuse of antimicrobials in developing countries in the first place uncalled for, resulting in unnecessary increase in AMR. The authors should try to bring out this fact also in the discussion section / limitation section so as to advocate rational use of antibiotics.

4. In view of the fact that studies with certain antibiotics taken in the panel e.g Meropenem (Carbapenams), 2nd and 3rd generation Cephalosporins etc are extremely low (1 or 2 only) sweeping statements on not finding resistance to these antibiotics may not be appropriate (only a few of the total isolates were actually tested against these antibiotics).

5. It would have been better to provide trends in antimicrobial resistances in various pathogens that would help clinicians for their choice of empirical treatment, when absolutely required.

There are other typos and small grammatical errors e.g many words have got joined together (might be due to PDF conversion or ununiform versions of MSO). There is too much use of the word 'causes' in Introduction lines 68-72; Line 164 after Figure 2, Almost all studies (97.67%) ere institutional-based... may be replaced with 'institution-based'; Line 233 'Resistant against...' may be replaced with 'Resistance against...'.

---

## [Author Response · Author response to Decision Letter 0]

18 Feb 2022

The responses are addressed point by point and attached.

---

## [Editor Report · Decision Letter 1]

28 Feb 2022

Prevalence and drug resistance patterns of Gram-negative enteric bacterial pathogens from diarrheic patients in Ethiopia: A systematic review and meta-analysis

PONE-D-21-39208R1

Dear Dr. Beyene,

We’re pleased to inform you that your manuscript has been judged scientifically suitable for publication and will be formally accepted for publication once it meets all outstanding technical requirements.

Kind regards,

Simon Clegg, PhD

Academic Editor

PLOS ONE

Additional Editor Comments:

Many thanks for resubmitting your manuscript to PLOS One

As you have addressed all the comments and the manuscript reads well, I have recommended it for publication

You should hear from the Editorial Office shortly.

It was a pleasure working with you and I wish you the best of luck for your future research

Hope you are keeping safe and well in these difficult times

Thanks

Simon

---

## [Editor Report · Acceptance letter]

8 Mar 2022

PONE-D-21-39208R1 

Prevalence and drug resistance patterns of Gram-negative enteric bacterial pathogens from diarrheic patients in Ethiopia: A systematic review and meta-analysis 

Dear Dr. Beyene:

I'm pleased to inform you that your manuscript has been deemed suitable for publication in PLOS ONE. Congratulations! Your manuscript is now with our production department. 

Kind regards, 

on behalf of

Dr. Simon Clegg 

Academic Editor

PLOS ONE